# The Future of the COVID-19 Pandemic: How Good (or Bad) Can the SARS-CoV2 Spike Protein Get?

**DOI:** 10.3390/cells11050855

**Published:** 2022-03-02

**Authors:** Matthew A. Nugent

**Affiliations:** Department of Biological Sciences, University of Massachusetts Lowell, Lowell, MA 01854, USA; matthew_nugent@uml.edu

**Keywords:** ACE2, angiotensin converting enzyme 2, co-receptors, coronavirus, heparan sulfate proteoglycans, heparin, HSPGs, Omicron, receptors

## Abstract

Severe acute respiratory syndrome virus 2 (SARS-CoV2) has infected an estimated 400 million people world-wide, causing approximately 6 million deaths from severe coronavirus disease 2019 (COVID-19). The SARS-CoV2 Spike protein plays a critical role in viral attachment and entry into host cells. The recent emergence of highly transmissible variants of SARS-CoV2 has been linked to mutations in Spike. This review provides an overview of the structure and function of Spike and describes the factors that impact Spike’s ability to mediate viral infection as well as the potential limits to how good (or bad) Spike protein can become. Proposed here is a framework that considers the processes of Spike-mediated SARS-CoV2 attachment, dissociation, and cell entry where the role of Spike, from the standpoint of the virus, is to maximize cell entry with each viral-cell collision. Key parameters are identified that will be needed to develop models to identify mechanisms that new Spike variants might exploit to enhance viral transmission. In particular, the importance of considering secondary co-receptors for Spike, such as heparan sulfate proteoglycans is discussed. Accurate models of Spike-cell interactions could contribute to the development of new therapies in advance of the emergence of new highly transmissible SARS-CoV2 variants.

## 1. Introduction

Severe acute respiratory syndrome virus 2 (SARS-CoV2) has infected an estimated 400 million people world-wide, causing approximately 6 million deaths from severe coronavirus disease 2019 (COVID-19) (https://covid19.who.int, accessed on 1 March 2022). Each infection presents multiple opportunities for mutation during the replication of the SARS-CoV2 genome that have the potential to, in evolutionary terms, enhance the “fitness” of a new variant leading to increased transmission among its human hosts. Indeed, the world has already experienced several major waves of infection with the rise of the highly contagious and virulent Delta variant (B.1.1.617.2) in the late summer of 2021, and now with the emergence of the even more highly transmissible Omicron variant (B.1.1.529). Thankfully, the wide distribution of effective vaccines coupled with the observation that the Omicron variant appears to cause less severe disease has led to a significantly reduced case-fatality rate with the Omicron variant compared to Delta. Nevertheless, the high rate of infection with Omicron presents the possibility that new, more virulent strains of SARS-CoV2 will develop, which raises the question of whether there is a limit to how infectious this virus can become. While the mechanism of viral infection involves a vast number of variables, the SARS-CoV2 Spike protein is one well-defined component that plays a critical role in viral attachment and entry into host cells [1]. Therefore, it is of interest to consider the structural and functional limits of Spike as we consider and prepare for future developments in the COVID-19 pandemic.

This review will focus on the factors that impact Spike protein function and will discuss the theoretical limits to how good (or bad) Spike protein can become.

## 2. Spike Protein Structure and Function

Spike is a ~180 kDa protein that protrudes from the surface of the SARS-CoV2 particle where it mediates host-cell attachment and entry [1,2,3,4,5]. The SARS-CoV2 Spike is expressed as a trimer on the SARS-CoV2 surface, with approximately 25 Spike trimers per virus particle [6]. The human transmembrane glycoprotein angiotensin converting enzyme 2 (ACE2) has been identified as the cognate receptor for Spike [7,8,9,10]. Furthermore, ACE2 is expressed in various tissues, including prominently within type I and type II alveolar epithelial cells in the lungs, and in nasal and bronchial epithelium [11,12]. Although ACE2 expression within the respiratory system is consistent with the pathology of COVID-19, it is interesting to note that ACE2 is also highly expressed in oral epithelia, enterocytes within the small intestine and the vascular endothelium. While the broad expression pattern of ACE2 indicated that viral disease could spread through many routes, it has become clear that respiratory transmission is the dominant means. Hence, expression of ACE2 seems necessary for infection, yet it does not, itself, appear to be a sufficient predictor of which cells will be most susceptible to SARS-CoV2.

The binding of Spike to ACE2 positions Spike on the cell surface for cleavage occurs by human proteases, particularly transmembrane protease serine 2 (TMPRSS2), that induces a conformational change from a compact structure to an extended “fusion” structure that facilitates the fusion of the viral and host cell membranes, thereby allowing viral RNA to gain entry to the cellular cytoplasm where it can be translated and replicated to produce new viral particles (Figure 1). The ability of TMPRSS2 to induce the conversion of Spike into the fusion conformation is enhanced if Spike is first cleaved by the protease furin during biosynthesis [13,14,15]. Thus, the expression and distribution of ACE2 determines the tissues that are infected by SARS-CoV2 and ultimately mediates COVID-19 pathogenesis. Interesting variations in ACE2 expression have been reported in individuals by age, gender, and disease, suggesting potential explanations for the highly variable clinical responses to SARS-CoV2 infection [16,17,18].

Spike protein from wild-type SARS-CoV2 contains 1273 amino acid residues arranged into two subunits, S1 (residues 14-685) and S2 (residues 686-1273) [1,3,5,19]. The receptor binding domain (RBD) is within the S1 subunit (residues 319-541) and contains the region that interacts directly with ACE2, called the receptor binding motif (RBM) (residues 437-507). The protease cleavage site responsible for the transition of Spike into the “fusion” conformation are within the S2 subunit. The SARS-CoV2 Spike protein also contains a unique insertion of a furin cleavage site, PRRA, at the S1/S2 junction that allows furin to prime Spike during biosynthesis for future cleavage by TMPRRS2 (or cathepsin in late endosomes). Thus, considerable attention has been focused on the particular characteristics of the Spike RBD and furin cleavage site to gain insight into how SARS-CoV2 has become so highly transmissible among the human population.

Spike protein is heavily glycosylated, with the exception of the RBD, providing extensive shielding from antibody recognition [20,21,22,23] (Figure 2). Interestingly, the RBD is structurally flexible and can dynamically switch between an open (up) and closed (down) conformation with the closed conformation being the preferred state. Spike can only bind to ACE2 when its RBD is in the open conformation. Within a given trimeric Spike, the three RBDs dynamically interact to stabilize the closed state such that it is rare to find more than one RBD in the open state at a time. Indeed, a Cryo-EM study of Spike protein indicates that, on average, an RBD will be in the open conformation in only ~20% of Spike trimers, while 80% will have all three RBDs in the closed state [3,24]. Thus, the relative amount of time that the RBD spends in the open state increases the probability that an encounter between Spike and ACE2 will lead to the formation of a bound complex while also increasing the chance that Spike will be recognized by host antibodies. Hence, Spike has likely evolved this conformational dynamic process to balance between these two competing events as a means of enhancing its ability to infect. Interestingly, the glycosylation of Spike also participates actively in the transition between the open versus closed state of the RBD [25,26]. In a unique mechanism, glycans at N165 and N234 stabilize RBD when it is in the open conformation while the glycan at N343 physically pushes the RBD up into the open conformation. Mutating any of these three residues reduces binding of Spike to ACE2 with the loss of the glycan at N343 causing a greater than 50% reduction in binding [26].

Since the original identification of SARS-CoV2, a number of variant forms have evolved and spread. Each of the variants show mutations in the Spike protein that have been linked to altered virulence and/or transmissibility. The Alpha (B.1.1.7), Beta (B.1.351) and Gamma (P.1) variants all carry an N501Y mutation within the RBM that has been demonstrated to increase the binding affinity to ACE2 by 3- to 7-fold, depending on the presence of additional mutations [27,28,29]. The recent emergence of the highly contagious Delta (B.1.1.617.2) and Omicron (B.1.1.529) variants reveal a number of additional important mutations in Spike [30]. The Delta variant has two mutations in its RBM, L452R and T478K, and in some instances has additional mutations K417N and E484K. The Omicron variant Spike protein, in contrast, carries as many as 30 point mutations, three deletions and one insertion. Strikingly, it has 15 mutations within its RBD with 10 tightly clustered within the RBM, namely N440K, G446S, S477N, T478K, E484A, Q493K, G496S, Q498R, N501Y and Y505H (Figure 3). It is interesting to note that these 10 mutations result in a significant change in the charged nature of the RBM with the substitution of four non-charged residues (N, T, and 2Qs) with positively charged residues (R and 3Ks), the loss or one negatively charged residue (E) and the addition of one histidine which can carry a positive charge depending on its local environment. A systematic analysis of each single Omicron mutation on ACE2 binding revealed that nine of the RBD mutations would be expected to decrease binding affinity while the other six should increase affinity [31]. Indeed, a comparative analysis of WT, Delta and Omicron revealed that the Delta Spike binds with higher affinity compared to WT and Omicron. Interestingly, the Omicron Spike does not show increased affinity for ACE2 compared to WT Spike [31]. Hence, it appears that the binding affinity for ACE2 is not the sole determinant of the effectiveness of Spike protein that dictates the enhanced transmission of SARS-CoV2 variants.

## 3. Kinetic Considerations of Spike Function

The first critical step in the process of SARS-CoV2 infection is the binding of Spike proteins to ACE2 on the surface of target host cells. This binding can, as a first approximation, be considered a classic receptor–ligand interaction, in which ACE2 is the receptor (A) and Spike is the ligand (S) and the interaction is defined by its association and dissociation rate constants and corresponding equilibrium dissociation constant (Equations (1) and (2)). As such, the “effectiveness” of Spike would be expected to be controlled by the number of ACE2 receptors on the surface of the target cell and the intrinsic affinity of the interaction. However, this simplification does not consider the multivalency of the virus particle, the movement and restriction of ACE2 within the two-dimensional plasma membrane nor the fact that the virus can be removed from the system after membrane fusion and/or endocytosis. A more complete treatment of the interaction should include the diffusion of the virus particle on the membrane surface after primary collision with the cell, rebinding of Spike to ACE2, generation of multiple Spike-ACE2 complexes on a single viral particle, and non-reversible Spike cleavage and cell entry steps. While a formal treatment of this complex interaction is beyond the scope of this review, the reader is directed to several elegant kinetic models of virus binding that have been developed [32,33,34,35]. In particular, the Brownian adhesive dynamics model developed by English and Hammer [34], where viral attachment is dictated by a balance of thermal, hydrodynamic and adhesive forces, the lateral movement of receptors in the membrane, and the availability of viral attachment proteins, demonstrates how important it is to consider these additional aspects in predicting virus interactions with target cells. For example, allowing receptors to move freely within the membrane significantly reduced the dependence of virus attachment on receptor density, whereas the number of virus particles bound at steady-state was insensitive to increases in receptor binding on rate once it was greater than the rate of receptor diffusion.
(1)$$S+A\underset{k_{off}}{\overset{k_{on}}{⇄}}S•A$$
(2)$$K_{D}=\frac{\left[S][A\right]}{\left[S•A\right]}=\frac{k_{off}}{k_{on}}$$

Equation (1); Spike (S) and ACE2 (A) interact to form a stable bound complex (S•A) that is defined by rate constants (*k_on_*, *k_off_*). Equation (2); the relationship of the equilibrium concentrations of S, A and S•A to the equilibrium dissociation constant (K_D_) and the rate constants (*k_on_*, *k_off_*).

Proposed here is a model that considers the processes of SARS-CoV2 attachment, dissociation, and cell entry such that the role of the Spike-ACE2 interaction, from the standpoint of the virus, is to maximize cell entry with each viral-cell collision (Figure 4). Thus, once a virus particle has encountered a target cell there are two possible fates: the virus dissociates from the cell or gains entry. The evolution of the virus would invariably select for mutations that increase the likelihood that each collision leads to cell entry. Herein the Entry Probability (EP) is proposed to provide a simple framework to describe this concept. The EP is simply the pseudo first order rate constant describing the cell entry process (*k_e_*) divided by the sum of *k_e_* and the pseudo first order rate constant describing release (*k_r_*) of the virus from the cell. Changes in Spike that decrease release (i.e., tighter binding to ACE2) or increase cell entry (i.e., more efficient transition to the “fusion” structure) will increase EP. An EP equal to 1 would indicate that every time a virus particle collides with the cell surface, it gains entry into the cell. To put this in similar terms, an EP of 1 would indicate that the cell attachment and entry functions of Spike have evolved to “perfection”.

Since the SARS-CoV2 Spike protein mediates cell attachment and entry, the above treatment raises the question, in what ways could additional mutations in Spike improve the EP for SARS-CoV2? To formally address this question, a number of key parameters would be required, including the density of ACE2 and TMPRSS2 on cell surfaces and the various kinetic constants defining their interactions with Spike. A number of studies have characterized the relative expression of ACE2 at the mRNA and protein level in various organs, tissues and cell types [12,36]. However, these data do not provide a means to estimate the receptor number per se. It is also necessary to determine the resident time of a virus particle on the cell surface in the absence of binding interactions (i.e., Step 1 in Figure 4). Several studies have measured and/or simulated virus movement on cell surfaces, model membranes, and through various media [32]. Based on these data, the area of the cell surface that a virus particle might scan prior to either engaging a receptor or dissociating from the cell surface is estimated to be in the range of 0.1–1 µm^2^. A reasonable estimate of ACE2 density, based on values for other receptors, is 1000–100,000 per cell [37]. Thus, the surface of a large cell, such as an alveolar type I epithelial cell, might present ~1–15 ACE2 proteins per µm^2^, assuming that ACE2 is uniformly distributed on the plasma membrane. As such, each SARS-CoV2 collision with a type I cell might result in relatively few Spike-ACE2 encounters, which, based on the kinetics of Spike-ACE2 binding [29], would be unlikely to lead to a stable, bound complex after most virus-cell collisions. However, it is important to note that studies have observed a non-uniform distribution of ACE2 whereby some cells in a population show high expression and others very low. Thus, it is possible that there is a sub-set of cells that are highly susceptible to SARS-CoV2 infection. Nevertheless, this rough analysis suggests that ACE2 levels may be limiting such that continued evolution of Spike that leads to increased binding affinity could translate to an increase in the EP.

Tracking individual fluorescently labeled virus particles on cell surfaces reveals a complex process of heterogeneous movements [38,39]. Virus particles are observed to undergo three types of movements: random diffusion, retrograde drifts, and confined motion which have been attributed to interactions between the virus and membrane proteins, the cortical actin matrix, as well as specific subdomains of the plasma membrane such as lipid rafts. Therefore, to more fully estimate the process by which SARS-CoV2 interacts with target cells, it is critical to consider potential sites of interaction between Spike and the cell surface in addition to ACE2. While the evidence that ACE2 is required for SARS-CoV2 to infect cells is compelling, there is also considerable data implicating other secondary binding sites, such as heparan sulfate proteoglycans (HSPGs). HSPGs have been well-characterized as high-density cell surface co-receptors that allow cells to increase their sensitivity to a wide-array of growth factors and cytokines [40,41,42]. These additional binding sites may have a significant impact on the ability of cells to capture SARS-CoV2 and increase the likelihood of Spike-ACE2 binding and eventual cell entry.

## 4. Heparan Sulfate Proteoglycans and Spike Interactions

HSPGs are a class of macromolecules expressed on cell surfaces and within the extracellular matrix of nearly all mammalian cells and tissues [43,44,45]. HSPGs are characterized by a core protein with covalently attached heparan sulfate (HS) chains. HS is a class of negatively charged linear polysaccharides characterized by repeating disaccharide units of alternating *N*-substituted glucosamine and hexuronic acid residues subject to selective modification including sulfation of the *N*-position as well as the *C-6* and *C-3 O*-positions of the glucosamine and the *C-2 O*-position of the uronic acid. Thus, the 32 (or more) potential unique disaccharide units and their grouping into structural motifs make this class of compounds one of the most information-dense in biology, providing a wide-array of protein-binding sites [46,47,48]. As such, HSPGs have been shown to serve as co-receptors to assist the growth factor binding to its tyrosine kinase receptors [41,42,49], and have also been demonstrated to provide attachment and entry sites for a number of viruses [50,51]. The high density of HSPG expression on cell surfaces combined with the large flexible HS chains are features that make this class of molecule ideal as initial attachment sites. Consequently, proteins and viruses can use HSPG to scan large regions of the cell surface to increase the probability of encountering lower density specific receptors. Moreover, as has been well characterized for heparin-binding growth factors, the presence of distinct HS and receptor binding sites on proteins allow HS to contribute to the formation of high-affinity ternary complexes that enhance ligand-receptor binding and signaling [42,49].

Additionally, the SARS-CoV2 Spike protein has been shown to bind HS and heparin (a highly sulfated form of HS expressed by mast cells), which is thought to be reflective of Spike-HS interactions on the cell surface that promote virus entry [52,53,54,55,56]. Indeed, removal of cell surface HS with heparinase has been shown to reduce SARS-CoV2 infection in cell culture models [52,54]. Spike has also been shown to bind to ACE2 and heparin independently in vitro such that a ternary complex can be generated involving all three molecules [53]. Computational docking and molecular dynamics simulations have identified a putative heparin/HS binding domain on Spike, a long positively charged patch that overlaps the S1/S2 junction and extends throughout the RBD [53,57]. Electron microscopic analysis indicates that heparin enhances the open conformation of the RBD suggesting a model whereby Spike binding to HS facilitates Spike-ACE2 interactions through a variety of mechanisms [53]. These findings, combined with the clinical observations that COVID-19 patients treated with heparin show improved outcomes [58,59], have led to a number of studies aimed at targeting Spike-HS interactions as a means of blocking SARS-CoV2 infection [60]. Most of these approaches are based on a model whereby soluble and/or extracellular heparin will bind to Spike protein and competitively inhibit binding to HS on the cell surface [56]. As such, there are more than 100 active clinical trials in the US exploring the use of heparin to treat COVID-19, including trials that are evaluating the delivery of heparin via nasal sprays or nebulizers to protect susceptible tissues against SARS-CoV2 infection (see: https://www.clinicaltrials.gov/ct2/results?cond=COVID-19&term=heparin&cntry=&state=&city=&dist=, accessed on 1 March 2022).

Considering the kinetic model described in Section 3, one might now consider the presence of HS, which generally presents protein-binding sites in the range of 10^6^–10^7^ per cell such that an initial collision of SARS-CoV2 with a cell might now be predicted to lead to hundreds to thousands of virus encounters with HS binding sites. Moreover, the binding of proteins to HS often show very fast kinetics such that proteins can bind, release, and rebind extensively, preventing their release from the cell surface [61,62]. This rapid binding mechanism provides a means for a protein, or a bound virus particle, to scan a large area of the cell surface to increase the chance of encountering and binding to lower density receptors. Thus, a SARS-CoV2 collision with the cell surface might be expected to result in Spike-HS interactions that ultimately facilitate the formation of stable HS-Spike-ACE2 complexes (Figure 5). Considering this HS co-receptor mechanism, the expression level and binding affinity of ACE2 on the cell surface might not be the key factors that limit SARS-CoV2 infection. Instead, the ability of Spike to interact with HS, or its susceptibility to proteolytic cleavage by TMPRSS2 might be the rate-determining factors.

In addition to HS, there are several reports indicating that Spike binds to other potential co-receptors such as neuropilin 1 and extracellular matrix metalloproteinase inducer (EMMPRIN, also known as basigin and CD147) [63,64,65]. Neuropilin 1 is a transmembrane glycoprotein that function as a co-receptor for semaphorins and vascular endothelial growth factor (VEGF) family members to regulate neurogenesis and angiogenesis [66]. Furthermore, neuropilin 1 has been demonstrated to collaborate with HS to form high affinity multimeric complexes with VEGF and its receptors [67]. Recently, neuropilin 1 was found to potentiate SARS-CoV2 infectivity using HEK-293T cells transiently expressing ACE2, TMPRSS2 and neuropilin 1. Furthermore, an analysis of human COVID-19 autopsies revealed that SARS-CoV2 infected neuropilin 1 positive cells in the nasal cavity [68].

EMMPRIN is an integral plasma membrane glycoprotein that is enriched on the surface of tumor cells where it stimulates the production of several matrix metalloproteinases by adjacent stromal cells [69,70]. EMMPRIN has also been implicated in oral cancers and inflammatory disorders, suggesting a connection to sites of SARS-CoV2 infection. Blocking EMMPRIN with antibodies or through knockdown in Vero E6 cells has been shown to inhibit SARS-CoV2 amplification while the expression of EMMPRIN in non-susceptible BHK-21 cells allows virus entry [65]. In addition, transgenic mice expressing human EMMPRIN showed high viral loads in the lungs, while wild-type mice did not. However, there remains some debate over whether direct Spike interactions with EMMPRIN are responsible for the observed enhanced SARS-CoV2 cell entry [71,72]. Nevertheless, the characterization of secondary Spike binding sites, in addition to ACE2, is revealing a broad range of mechanisms that Spike might employ to mediate SARS-CoV2 infection of cells.

The evidence that HS, neuropilin-1 and EMMPRIN are critical mediators of SARS-CoV2 infectivity via the ability to bind Spike and, in the case of HS, by stabilizing Spike-ACE2 interactions, necessitates that the models presented in Figure 4 and Figure 5 be expanded to capture the full nature of the viral-cell interaction process. As a first approximation, the ideas and models presented here have considered the fate of an individual virus particle interacting with a cell, with the assumption that these events would scale to be reflective of the larger population of virus particles. However, it is also important to begin to consider how interactions and competition between multiple virus particles on a given cell surface with the myriad of binding sites would impact cellular infection. Clearly this is a complex process, yet careful model-building coupled with quantitative experiments has the potential to identify components of the system that future viral variants might exploit to increase transmissibility. For example, accurate models could reveal that infectivity is sensitive to the affinity of Spike for HS but insensitive to that for ACE2 or vice versa. Consequently, this insight would help focus the development of drugs/interventions to combat future virus strains where they are likely to be vulnerable.

## 5. The Evolution of “Perfect” Spike Protein

The field of enzymology has considered the theoretical maximal rates that an enzyme can catalyze chemical reactions. Thus, a “perfect” enzyme has generally been defined as an enzyme whose catalytic efficiency has reach a theoretical maximum such that reaction is limited only by diffusion of the substrate to the enzyme [73]. As such, all collisions of substrate with a “perfect” enzyme lead to a conversation to product (i.e., there are no no-productive enzyme-substrate interactions). This concept applies to simple cases where enzymes and substrates move freely in solution and random diffusion is the only means for the molecules to collide. Interestingly, there are examples where enzymes catalyze reactions at rates that exceed the diffusion limitations (i.e., catalase). These “more perfect” cases generally act through the use of electrostatic forces whereby charged residues on the enzyme attract oppositely charged substrates, or through other means whereby scaffolds or other microdomains are created to concentrate enzyme or substrate in close proximity. Nevertheless, it can be valuable to consider the physical limits of a protein’s or enzyme’s function.

In considering the SARS-CoV2 Spike protein the obvious factors are the ability to bind to ACE2 and to be converted to the “fusion” conformation in order to mediate viral entry. Thus, when a large number of mutations were identified in the RBD and RBM of Spike protein from the highly transmissible Omicron variant, it was assumed that increased ACE2 binding was responsible for the increased transmission rate. Interestingly, direct measurement of Spike binding to ACE2 did not reveal an increase in affinity for Omicron compared to wild-type Spike. In fact, Omicron Spike showed reduced affinity compared to that from the less transmissible Delta variant. However, as noted above, the ability of Spike to encounter and form a tight complex with ACE2 appears to be mediated by HSPG co-receptors, thus, it is possible that Omicron Spike binding to HS or to ACE2 in the presence of HS would show interesting distinctions compared to other forms of Spike. Indeed, using in silico docking approaches, Clausen et al. [53] present evidence for electrostatic interactions between heparin and several basic residues within the RBD of wild-type Spike (R346, R355, K444, R466). In another molecular modeling study, potential interactions between heparin and several basic residues within the RBD and the S1/S2 site were also identified [57]. Consequently, the introduction of several basic residues within the RBD of Omicron Spike may potentially increase HS binding. Therefore, it is critical to characterize the secondary binding events in order to identify the mechanisms used by variant forms of Spike to increase transmission of SARS-CoV2.

HS has also been demonstrated to increase the propensity of the Spike RBD to be in the open conformation, where it is available to bind to ACE2, which would translate to an increased probability that a Spike encounter with ACE2 would lead to binding. Interestingly, a recent study of Spike protein with a mutation that is found in the Delta variant (D614G) showed a significant increase in the percentage of Spike trimers with one or two RBD in the open state [74]. Therefore, mutations in Spike that modulate the dynamics of the RBD, either directly or through changes in interactions with co-receptors could also increase cell-infection rates (i.e., decrease *k_r_* in Figure 4B).

In addition to the mutations characterized in the RBD in Spike variants, the Alpha Delta and Omicron Spike proteins all contain a proline to arginine substitution at position 681 that has been reported to improve the efficiency of furin cleavage. In these variants a greater percentage of Spike proteins will be primed for TMPRSS2 cleavage and subsequent cell fusion. Interestingly, Omicron Spike contains an additional histidine to tyrosine substitution at position 655 near the furin cleavage site. It is not yet known whether this additional mutation influences furin activity toward Spike. Nevertheless, possible future variants might act to improve furin, and/or TMPRSSS cleavage to enhance the cell entry process (i.e., increase *k_e_* in Figure 4B).

At this time, there are too many unknowns to accurately predict how far Spike is from perfection. Useful models will require accurate measurements of the number of ACE2 and TMPRSS2 molecules and HS binding sites per cell, as well as the binding and catalytic properties of TMPRSS2 toward furin-cleaved and non-cleaved Spike. Estimates of the relative fraction of Spike proteins on the virus that have been pre-processed by furin would also be critical. Nevertheless, the framework presented here may be helpful in providing a guide for future studies that aim to generate a more comprehensive understanding of the function of Spike. Models that help to identify processes that new variants might exploit to enhance viral transmission could provide insight for the development of new therapies in advance of the emergence of new highly transmissible variants.

## 6. Conclusions

Mutations in the Spike protein of SARS-CoV2 have been attributed to the evolution of highly transmissible variants that have led to waves of increased COVID-19 cases, hospitalizations and deaths. Each additional infection provides opportunity for additional mutations in Spike and other viral proteins that may lead to new variants that could elude vaccine-induced or natural immunity. This review has identified elements of Spike protein functions and additional relevant interactions that contribute to the ability of the virus to gain entry into target cells. The answer to the question of how much better (or worse in terms of causing human disease) Spike can become remains open, yet it is important to recognize that there is a limit. The concepts presented here may help guide studies to identify the aspects of viral infection that are subject to evolutionary improvement. Future viral pandemics are inevitable. Thus, it is critical that the scientific, public health and medical communities work together to foresee what evolution may have in store so that effective vaccines and therapies will be ready.

## Figures and Tables

**Figure 1 cells-11-00855-f001:**
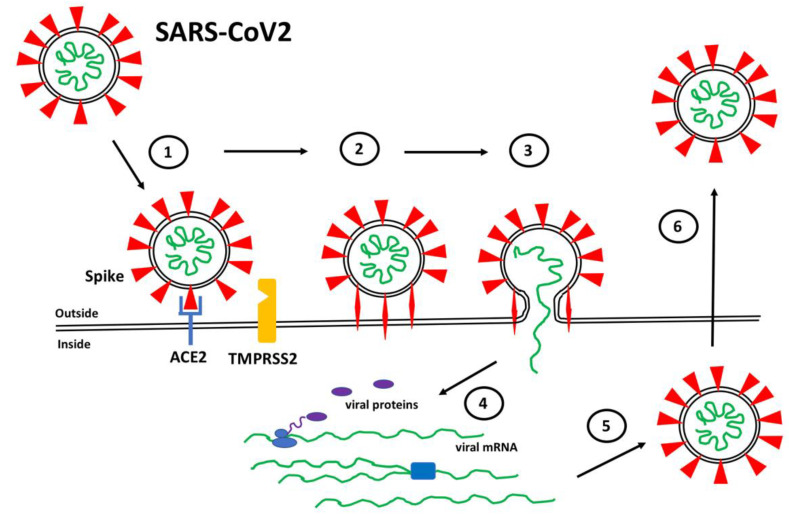
SARS-CoV2 Infection. (1) Spike proteins (red) on the SARS-CoV2 surface bind to ACE2 (blue) on the surface of target host cells. (2) Spike-ACE2 interactions position Spike for proteolytic processing on the cell surface by TMPRSS2 (orange) (or by cathepsin in endosomes after internalization). Cleavage of Spike releases a structural constraint that allows Spike to convert to its “fusion” structure that penetrates the host cell membrane. (3) “Fusion” Spike facilitates fusion between the viral and host membranes, allowing entry of viral RNA. (4) Viral RNA is translated and replicated within the host cell. (5) New virus particles are packaged in the host cell where furin cleaves Spike so as to prime it for later conversion to the “fusion” structure. (6) New SARS-CoV2 particles are released (or transferred to adjacent fused cells) to propagate further host cell infection.

**Figure 2 cells-11-00855-f002:**
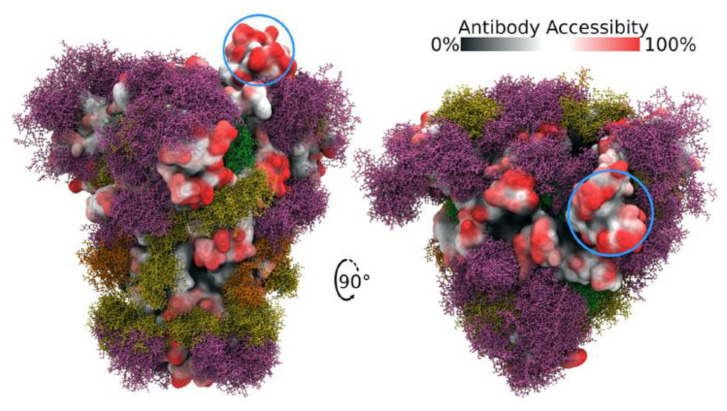
Side and top views of Spike trimer with site-specific glycosylation shown (moss surface) from molecular dynamics simulations [23]. The glycans are shown in ball-and-stick representation in green, dark yellow, orange and pink. The protein surface is colored according to antibody accessibility from black (least) to red (most accessible). One RBD is shown in the open conformation (circled in blue). Images are from [23] and are used under Creative Commons Attribution 4.0 International License: https://creativecommons.org/licenses/by/4.0/, accessed on 1 March 2022.

**Figure 3 cells-11-00855-f003:**
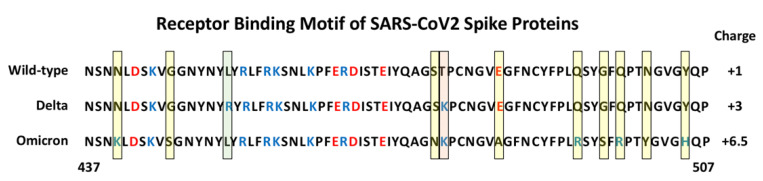
Amino Acid sequence of the Receptor Binding Motif of Spike Proteins. The sequences from position 437 to 507 are shown for the wild-type, and Delta and Omicron variants. The yellow shaded boxes indicate unique mutations in Omicron, the green box indicates a unique mutation in Delta, the pink box indicates a mutation shared by Delta and Omicron. Basic amino acids are colored blue and acidic amino acids red. The estimated charge for each RBM is indicated on the right (+0.5 was assigned to H505 in Omicron Spike).

**Figure 4 cells-11-00855-f004:**
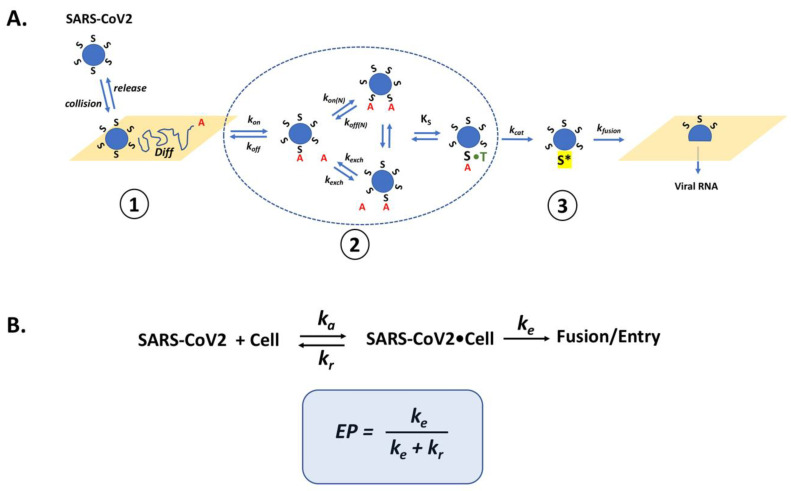
Kinetic Model of SARS-CoV2 Cell Interactions. (A) The model is presented as the composite of three general processes: (1) Initial collision and release from the cell surface (represented by the yellow plane) coupled to random diffusion. (2) Reversible binding of Spike protein (S) to ACE2 (**A**) (defined by kinetic constants *k_o_*_n_ and *k_off_*) and to TMPRSS2 (T) represented by the substrate binding constant (Ks). These events include the rebinding/exchange of one S•A complex with another (*k_exch_*) as well as the recruitment of additional S•A complexes (*N*) represented by *k_on(N)_/k_off(N)_*. (3) The irreversible proteolytic cleavage of Spike by T to its fusion form S*, represented by *k_cat_*, and the fusion of Spike with the cell membrane (*k_fusion_*). Additional terms can be added to describe the nonreversible endocytosis of SARS-CoV2 and cleavage of Spike by cathepsin in endosomal compartments. (**B**) The Entry Probability (EP) describes the likelihood that a SARS-CoV2 particle will enter a cell after it has collided with a target cell based on a simple kinetic model describing the lumped processes of virus association (*k_a_*) and release (*k_r_*) from the cell as well as fusion and entry (*k_e_*). Thus, the EP is based on the ratio of the pseudo first order rate constant of SARS-CoV2 entry into the cell (*k_e_*) after Spike-mediated fusion (Process (3)) and virus release from the cell surface (*k_r_*) either before or after various S•A and S•T complexes have formed (combination of Processes (1) and (2)).

**Figure 5 cells-11-00855-f005:**
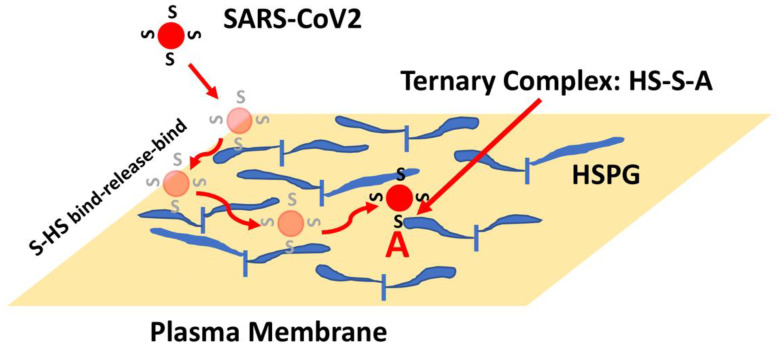
HSPG-mediated Binding of Spike Leads to Increased Binding to ACE2. The initial collision of SARS-CoV2 with the cell surface is likely to result in encounters with high-density and large volume HS chains. Binding of Spike (S) to HS can result in binding, release and re-binding to adjacent HS chains allowing for viral particle movement from one HS chain to another until Spike engages ACE2 (A). Spike-ACE2 interactions are stabilized by the combined binding of HS within a ternary complex.

## Data Availability

Not applicable.

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
