# Peer review of "The Future of the COVID-19 Pandemic: How Good (or Bad) Can the SARS-CoV2 Spike Protein Get?"

_cells, 2022, doi:10.3390/cells11050855_

Round 1

Reviewer 1 Report

This review brings important information about Spike protein from SARS-CoV-2 and identified elements of Spike function and additional relevant interactions that contribute to the ability of the virus to be more infective accordingly with the amino acids mutations and the variants spreading.  

Suggestion: Figure 1 should be drawn using software to improve the image quality.

Author Response

Thank you for the comments and suggestion to re-draw Figure 1. We have prepared a new Figure 1 using software to improve the image quality and clarity. 

Reviewer 2 Report

Dear All,  

The review manuscript entitled "The Future of the COVID-19 Pandemic: How Good (or Bad) Can the 2 SARS-CoV2 Spike Protein Get?" by Matthew A. Nugent, aims to provide an overview of the structure and function of SARS-CoV2 Spike protein and a description of those factors that may impact the Spike´s ability to mediate viral infection.The author outline some of the processes of Spike-mediated SARS-CoV2 attachment, dissociation, and cell entry, focusing on the factors that affect the protein function an describe the theoretical limits of how good (or bad) the Spike protein can become.

Some relevant aspects of the Spike Protein Structure and Function are mentioned. Additionally,   Kinetic Considerations of Spike Function were considered and a  Kinetic Model of SARS-CoV2 Cell Interactions is presented. In my opinion this is an interesting contribution.  Subsequently, a review of the potential participation of several co-receptors is presented and an expansion of the previous model to capture the full nature of the viral-cell interaction process is suggested.

Although it presents an interesting perspective, some aspects needs improvement for clarification.

I recommend the publication of the manuscript in your journal only after addressing the following points:

1.-  In section 2 some aspects of the extensive shielding of the Spike protein by glycans are discussed.  It has been reported an abrupt decrease in glycan shielding when RBD transitions from the "down" to the "up" state. N165 and N234 glycans consistently shield the RBM, while shielding by N343 glycan decreases by RBD opening. N165 and N234 glycans have a structural role stabilizing the "up" conformation, while N343 glycan acts as a "glycan gate" pushing the RBD from "down" to "up" conformation.  In my opinion this should be mentioned and take into account (Sztain et al. https://doi.org/10.1038/s41557-021-00758-3; Casalino et al. doi:10.1021/acscentsci.0c01056).

2.-  A sequence alignment of the RBD of the variants showing mutations in the Spike protein would be helpful to improve clarity.

3.- Figure 1 should be designed in accordance with the rest of the figures in the article.

4.- Figure  5 is of low quality and does not provide any additional or relevant information. The figure can be removed

5.- In section 4 a figure of the SARS-Cov-2 Spike RBD that highlight the interaction of the different co-receptors would be helpful

6.- In line 377 " it is possible that Omicron Spike binding to HS or to ACE2  in the presence of HS would show interesting distinctions compared to other forms of  Spike. Indeed, as noted above, the introduction of several basic residues within the RBD 379 of Omicron Spike are likely to increase HS binding"  to support this statement the SARS-CoV-2 spike protein with heparan sulfate through its receptor-binding domain (RBD) must be better described in section 4.

Author Response

I thank the reviewer for their thorough review and helpful comments. I have addressed each point below and believe that the paper is now significantly improved as a result.

  1. (Active role of glycans in Spike structure, page 3) We have added discussion of the role of specific glycans in stabilizing and mediating the transition of the RBD of Spike from the down to the up position and have included the relevant references to this intriguing mechanism of control (see page 3 in the revised manuscript).

  1. (New Figure illustrating sequence alignments, page 4) We have included a new Figure 3 to show the sequence alignments of the RBMs of Spike from the original, the Delta and the Omicron strains.

  1. (Re-draw Figure 1, page 2) Figure 1 has been redrawn so as to be consistent with other figures in the paper and to improve quality and clarity.

  1. (Purpose of old Figure 5 (now Figure 6), page 8) The purpose of Figure 5 (now Figure 6) is to draw the reader’s attention to the potential role of HSPG in expanding the fraction of the cell surface that the virus can sample so as to increase the probability of contact between Spike and ACE2. HSPG also stabilize the Spike-ACE2 interaction via the formation of a ternary complex. I have modified the previous version of this figure and edited the legend to better illustrate these points.

  1. (Suggested new figure) Unlike with Spike and ACE2, the details of how Spike interacts with its co-receptors are not well defined. Thus, I am not able to prepare an accurate and informative figure showing these interactions that would significantly add to the review.

  1. (Statement regarding Omicorn Spike-HS interactions, page 10) We have added information from various studies that have implicated basic regions within the RBD and elsewhere in mediating heparin-Spike bidning. I have also changed the language of the statement regarding how the mutations might alter Spike-HS interactions so as to better represent the speculative nature of this comment (specifically I changed “likely” to “may potentially”).

Reviewer 3 Report

The review work presented by Matthew A. Nugent titled “The Future of the COVID-19 Pandemic: How Good (or Bad) Can the SARS-CoV2 Spike Protein Get” is well written, clear, and easy to read. The topic is very interesting and therefore, it adds clustered information to the subject area of infectious diseases mediated by coronavirus with a focus on the SARS-CoV2 infection. It is a cutting edge area and we still not have drug against the COVID-19 disease. In particular, the author performed a very well-conceived overview about the role of Spike viral protein its structure and function in viral entry via ACE2 evocating generally the cytokine storm mediated inflammation.

Original the Figure 1!

The section on Heparan Sulfate Proteoglycans and Spike Interactions have the generic link of clinical trial.gov might be it should be better to link directly the one related to heparin use.

Author Response

Thank you for your comments and suggestions. I have made the edits to address these as noted below.

Figure 1 has been redrawn so as to be consistent with other figures in the paper and to improve quality and clarity (see page 2).

I have added a hyperlink to the list of clinical trials evaluating the role of heparin or heparin-like agents for the treatment of COVID-19 (see page 8).

Round 2

Reviewer 2 Report

Dear all,

I recommend the publication of the manuscript in your journal in the present form.